# Richter Transformation in Chronic Lymphocytic Leukemia: Current Treatment Challenges and Evolving Therapies

**DOI:** 10.3390/ijms26178747

**Published:** 2025-09-08

**Authors:** Zi-Chi Lin, Ming-Jen Chan, Tang-Her Jaing, Tung-Liang Lin, Yu-Shin Hung, Yi-Jiun Su

**Affiliations:** 1Division of Hematology-Oncology, Department of Internal Medicine, Linkou Chang Gung Memorial Hospital, Taoyuan 333423, Taiwan; mp2750@cgmh.org.tw (Z.-C.L.); ldl2605@cgmh.org.tw (T.-L.L.); f22338@cgmh.org.tw (Y.-S.H.); 2Department of Nephrology, Kidney Research Center, Linkou Chang Gung Memorial Hospital, Taoyuan 333423, Taiwan; b9202066@cgmh.org.tw; 3College of Medicine, Chang Gung University, Taoyuan 333423, Taiwan; jaing001@cgmh.org.tw; 4Division of Hematology/Oncology, Department of Pediatrics, Chang Gung Memorial Hospital in Linkou, Taoyuan 333423, Taiwan

**Keywords:** Richter transformation, chronic lymphocytic leukemia, chemoimmunotherapy, CAR T-cell therapy, bispecific antibodies, BTK inhibitors, BCL-2 inhibitors, allogeneic hematopoietic stem cell transplantation

## Abstract

Richter transformation (RT) affects 2–10% of chronic lymphocytic leukemia (CLL) patients, evolving into an aggressive lymphoma—most often diffuse large B-cell lymphoma—with poor prognosis, especially when clonally related to CLL. Key risk factors include unmutated IGHV, *TP53* and *NOTCH*1 mutations, stereotyped B-cell receptors, and complex cytogenetics. This review summarizes RT biology, clinical predictors, and treatment outcomes. Traditional chemoimmunotherapy (e.g., R-CHOP) yields complete response rates around 20–30% and median overall survival of 6–12 months; intensified regimens (R-EPOCH, hyper-CVAD) offer only modest gains. Allogeneic hematopoietic stem cell transplantation is potentially curative but limited to fit patients due to high treatment-related mortality. Emerging therapies now include Bruton’s tyrosine kinase and BCL-2 inhibitors, which achieve partial responses but short progression-free survival. CD19-directed chimeric antigen receptor T-cell therapies produce overall response rates of 60–65%, though relapses remain frequent. Bispecific antibodies (e.g., CD3×CD20 agents epcoritamab and mosunetuzumab) show promising activity and tolerable toxicity in relapsed/refractory RT. Ongoing trials are exploring combinations with checkpoint inhibitors, triplet regimens, and novel targets such as ROR1, CD47, and CDK9. Continued research into optimized induction, consolidation, and innovative immunotherapies is essential to improve outcomes in this biologically distinct, high-risk CLL-related lymphoma.

## 1. Introduction

Chronic lymphocytic leukemia (CLL) is the most common adult leukemia in western countries, with an age-adjusted incidence of 4.6 per 1,000,000 individuals [1,2]. Richter transformation (RT) developed in 2 to 15% of CLL patients, with an incidence of 0.5 to 1% per year [3,4]. RT was first described by Maurice Richter in 1928, refers to the evolution of CLL into an aggressive lymphoma, most often diffuse large B-cell lymphoma (DLBCL), with occasional cases of Hodgkin lymphoma or rarer variants like B-lymphoblastic lymphoma and mantle cell lymphoma [5,6,7,8]. RT is characterized by rapidly progressive lymphadenopathies, the presence of B symptoms, elevated lactate dehydrogenase level, and frequent extranodal involvement [3]. The prognosis of RT is generally poor, particularly in cases of clonally related RT, which accounts for approximately 80 percent of all cases [9].

RT is predisposed by both clinical and biological high-risk features: lymphadenopathy > 5 cm, Rai stage III–IV, unmutated immunoglobulin heavy chain variable region (IGHV) status, *TP53* aberrations, *NOTCH1* mutations, presence of stereotyped B-cell receptors, over-expression of CD38 or ZAP-70, and a complex karyotype [10,11,12,13,14,15,16,17], commonly defined as the presence of ≥3 numeric or structural chromosomal abnormalities by conventional karyotyping [18] A Danish population study confirmed these features at scale, with an RT incidence of 4.5% and a higher risk in early-stage patients who later required treatment [19]. The median overall survival (OS) after a diagnosis of RT-DLBCL remains poor, at approximately 12 months [6]. Given this dismal prognosis, a deeper understanding of the biology and therapeutic landscape of RT is essential. This review aims to explore the molecular pathogenesis, clinical characteristics, and treatment outcomes of RT under various therapeutic strategies, including allogeneic hematopoietic stem cell transplantation (allo-HSCT), chimeric antigen receptor (CAR) T-cell therapy, bispecific antibody therapy, and ongoing or future clinical trials.

## 2. Molecular Features and Pathogenesis

RT can arise either from the original CLL clone, referred to as clonally related RT, or from a distinct B-cell clone, known as clonally unrelated RT [9,20]. Approximately 80% of RT-DLBCL cases are clonally related, retaining the same IGHV rearrangements as the CLL and reflecting true clonal evolution driven by the stepwise acquisition of high-risk genetic lesions [21]. In contrast, roughly 20% of cases are clonally unrelated, arising as de novo lymphomas that lack a clonal connection to the CLL and typically exhibit clinical behavior closer to sporadic DLBCL, with comparatively superior outcomes [22]. Longitudinal studies of paired CLL and RT samples have illustrated this evolutionary process: Nadeu et al. demonstrated that minor subclones harboring RT-associated mutations can be present years—sometimes nearly two decades—before overt transformation [23]. These latent high-risk subclones may subsequently expand into the dominant RT clone when selective pressures such as prior therapy or microenvironmental cues confer a survival advantage. Over time, multiple competing subclones may diversify in parallel, but ultimately one lineage accumulates a critical constellation of driver lesions and emerges as the dominant aggressive clone that clinically defines RT [23]. This “early seeding” model underscores that RT is not a sudden event but rather the culmination of long-standing clonal diversification and Darwinian competition, highlighting the evolutionary continuum from indolent CLL to aggressive lymphoma.

At molecular level, RT is underpinned by recurrent genetic alterations that disrupt tumor suppressor networks and deregulate key signaling pathways, thereby propelling the indolent CLL clone into a high-grade malignant state. Four lesions in particular—*TP53* inactivation, *CDKN2A/B* loss, *NOTCH1* activation, and *MYC* dysregulation—consistently emerge as central hallmarks of RT pathogenesis, forming a genetic foundation upon which indolent CLL evolves into aggressive lymphoma [21,24]. Importantly, each of these alterations not only drives malignant progression directly but also cooperated with aberrant B-cell receptor (BCR) signaling, a central axis of CLL biology and a critical force in the pathogenesis of RT [24]. Figure 1 schematically illustrates the clonal evolution from CLL to RT and associated therapeutic implications.

Disruption of the *TP53* pathway is the most common event in RT-DLBCL, observed in approximately 60–70% of cases [21,24]. *TP53* loss, through mutation or deletion of chromosome 17p, disables the DNA damage checkpoint and apoptotic machinery, enabling unchecked accumulation of genomic lesions. Importantly, *TP53* mutations are often detectable in the CLL clone prior to transformation, suggesting that *TP53-*deficient subclones are predisposed to evolve into RT [23]. The functional interaction between *TP53* loss and BCR signaling is particularly revealing. In CLL cells with intact checkpoints, BCR stimulation paradoxically induces both pro-proliferative signals and the expression of CDK inhibitors such as p21 and p16, leading to cell-cycle arrest in the absence of co-stimulatory cues. However, when *TP53* and *CDKN2A/B* are inactivated, these braking mechanisms are removed, converting BCR engagement from an anergic to a proliferative signal [25]. Experimental models confirm that CLL cells with combined *TP53*/*CDKN2A* inactivation proliferative autonomously upon BCR stimulation and that this proliferation remains strictly BCR-dependent. Thus, *TP53* loss not only destabilized the genome but also rewires BCR signals into an engine of unrestrained clonal expansion.

Alongside *TP53*, deletion in the *CDKN2A/B* locus (chromosome 9p21) are observed in 20 to 30% RT cases [24]. These genes encode inhibitors of CDK4/6, which regulate the G1-S cell-cycle checkpoint. Their synergy with BCR signaling is profound: in the context of *CDKN2A/B* deletion, antigen-driven BCR activation directly propels cell-cycle entry without the counterbalance of inhibitory regulators. Functionally, this explained why RT cases with dual *TP53* and *CDKN2A/B* loss are particularly aggressive yet still reliant on BCR-derived signals, reflecting a paradoxical dependency even in highly genomically unstable clones [25]. This cooperative biology highlights the intersection of cell-cycle deregulation and chronic antigenic stimulation in RT pathogenesis.

Oncogenic driver mutations further cooperate with these tumor suppressor losses to drive the clonal evolution of RT. Activating mutations in *NOTCH1*, typically truncating mutations in the PEST domain, are present in 20 to 30% of RT-DLBCL [21]. These mutations result in persistent *NOTCH1* signaling and transcription of survival-promoting genes. Clinically, *NOTCH1* mutations are a well-established risk factor for RT: up to 45% of CLL patients harboring *NOTCH1* mutations eventually transform [24]. Importantly, *NOTCH1* signaling directly amplifies BCR downstream cascades, particularly PI3K-AKT and NF-κB pathways, promoting proliferation and apoptosis resistance. This convergence fosters a pro-survival transcriptional program, while also reducing reliance on external microenvironmental support. *NOTCH1* mutations are more frequent in CLL patients on BTK inhibitors, suggesting that therapeutic blockade of proximal BCR kinases may select for clones in which *NOTCH1* substitutes as an alternative amplifier of PI3K/AKT signaling. Thus, *NOTCH1* lesion exemplifies how RT biology adapts by sustaining oncogenic BCR outputs through parallel pathways [22].

Another major driver is dysregulation of *MYC,* a master oncogene that controls cellular proliferation and metabolism. *MYC* abnormalities are identified in 30 to 50% of RT cases, occurring via various mechanisms including gene amplifications, chromosomal translocations, or loss-of-function mutations in negative regulators such as MGA [24]. *MYC* activation drives metabolic reprogramming, cell-cycle acceleration, and transcriptional programs favoring aggressive growth. In RT, *MYC* alterations often co-occur with *TP53* inactivation, producing highly proliferative and genomically unstable clones. This synergy is thought to underline the explosive clinical behavior of RT-DLBCL. *MYC* activity also integrates with BCR signaling: NF-κB and PI3K/AKT cascades downstream of BCR upregulate *MYC* expression, while *MYC* in turn enhances glycolysis and oxidative phosphorylation, meeting the energetic demands of chronic signaling. Transcriptomic profiling of RT cells demonstrates enrichment of oxidative phosphorylation programs compared with parent CLL, consistent with *MYC*-driven metabolic rewiring layered upon BCR stimulation. This cooperation explains why *MYC*-aberrant RT clones often dominate the evolutionary landscape—they are not only genetically unstable but metabolically optimized for survival in the context of chronic antigenic signaling.

Beyond single-gene lesions, RT is distinguished by profound genomic instability. Whole-genome and exome sequencing have revealed RT harbors significantly more mutations, copy number alterations, and structural rearrangements than the antecedent CLL [21]. Catastrophic genomic events such as chromothripsis and chromoplexy are frequent, and around 15% of RT-DLBCLs exhibit whole-genome doubling, a phenomenon rarely seen in untreated CLL [21]. These phenomena reflect both the disabling of DNA damage checkpoints (via *TP53*/ATM defects) and the proliferative drive imposed by BCR and *MYC*. Intrinsic processes such as aberrant somatic hypermutation and telomere erosion further fuel mutagenesis, while extrinsic factors—including therapy-induced stress and microenvironmental signals—likely accelerate genomic chaos [24]. Ultimately, Darwinian selection among these genomically diverse subclones yields one dominant lineage bearing the optimal constellation of BCR signaling capacity, checkpoint disruption, and proliferative advantage.

The immunogenetic context of BCR signaling further supports its role in RT. Clonally related RT frequently arises from CLL clones with unmutated IGHV genes and stereotyped BCRs, particularly subset #8, which recognizes autoantigens such as vimentin and delivers strong signaling outputs [21]. Such clones are strongly enriched among patients who progress to RT, highlighting the contribution of chronic antigenic drive. High ZAP-70 expression in these CLL cells lowers the threshold for BCR activation, enhancing downstream cascades. In contrast, clonally unrelated RT, which resembles de novo germinal-center DLBCL, typically arises from mutated IGHV cells and lacks stereotyped BCRs, underscoring distinct pathogenic trajectories.

Morphologically, RT-DLBCL predominantly exhibits an activated B-cell or non-germinal center phenotype, consistent with its dependence on chronic active BCR signaling. Many cases retain CLL immunophenotypic features, including CD5 expression. Compared with de novo DLBCL, RT more frequently expresses PD-1/PD-L1, facilitating immune evasion. Integrative analyses now define molecular subtypes of RT based on driver combinations: *NOTCH1*/trisomy 12 and *SF3B1/EGR2*-mutant subtypes show somewhat lower genomic complexity, while *TP53*-deficient clones with whole-genome doubling display the greatest instability and poorest outcomes [21]. However, these proposed classifications are based on limited cohorts, and their reproducibility across diverse populations remains uncertain.

Despite advances, significant challenges remain in delineating RT pathogenesis. Most genomic studies have focused on heavily pretreated CLL, raising questions about whether therapy accelerates particular evolutionary paths. The temporal order of driver lesions and their relative contributions to transformation versus general disease progression remain incompletely defined, given the scarcity of prospective longitudinal data. Furthermore, while BCR signaling clearly plays a central role, the extent to which transformation is driven by intrinsic antigenic stimulation versus microenvironmental support is not fully resolved. Finally, functional insights largely stem from in vitro or murine models, which cannot fully recapitulate human CLL evolution or the stromal context of transformation. Improved longitudinal sampling, single-cell multi-omics, and patient-derived models will be critical to address these gaps.

In summary, RT represents the endpoint of prolonged clonal evolution in CLL, characterized by interplay between chronic BCR signaling and recurrent driver lesions. *TP53* inactivation, *CDKN2A/B* loss, *NOTCH1* activation, and *MYC* dysregulation converge with hyperactive BCR pathways to dismantle tumor-suppressor programs, remodel metabolism, and promote genomic instability. These synergistic events forge the dominant lymphoma clone, culminating in the aggressive and often treatment-refractory biology of RT.

## 3. RT Incidence and Outcomes

High-risk genomic alterations in CLL, including *TP53* aberrations, *NOTCH1* mutations, complex karyotype, and unmutated IGHV, are consistently associated with increased RT risk [10,11,12,13,14,15,16,17]. These factors, together with clinical features and prior therapy, support risk-adapted treatment planning and trial referral in RT [3,26].

Reported RT incidence varies widely depending on treatment era, patient population, diagnostic criteria, and follow-up duration. A summary of major studies and cohorts is shown in Table 1.

Differences in incidence likely reflect patient selection, biopsy practices, treatment exposure, and biological effects of therapy. Hypothesized mechanisms include selection of pre-existing aggressive subclones under therapeutic pressure, therapy-induced immune dysfunction, and microenvironmental changes that facilitate transformation. For example, prolonged BTK inhibition may promote outgrowth of *TP53*-mutated clones, while venetoclax may preferentially spare apoptosis-resistant populations.

Prospective data on RT are limited. A German CLL Study Group pooled analysis of frontline chemoimmunotherapy (CIT) trials reported RT in 3% of patients, with median OS of 9.4 months post-RT [28]. Earlier cohorts from MD Anderson and Mayo Clinic showed median OS of 8 and 7.8 months, respectively [6,27]. However, RT diagnosed in previously untreated CLL patients showed better outcomes with median survival up to 46.3 months, suggesting that treatment-naïve status at the time of transformation may be linked to less adverse outcomes [6,9].

Real-world registry data and clinical trial data support these findings. A Danish study found RT risk unchanged in the era of novel agents, with chemotherapy exposure linked to worse survival [19]. In RESONATE and RESONATE-2 trials, 4 to 5% of patients developed RT after ibrutinib [30,31]. Similar observations are reported in trials of venetoclax [32]. The Bruton’s tyrosine kinase inhibitor (BTKi) era brought hope of altering RT incidence and treatment. Frontline ibrutinib may reduce RT risk to 1 to 4% annually, but when progression occurs, it is often presented as RT [19,33,34]. Analyses from Ohio State and MD Anderson revealed that 30 to 40% of patients progressing on ibrutinib developed RT, with a dismal median OS of 3 to 6 months [34,35]. Collectively, these data suggest that although novel agents can delay CLL progression, they do not eliminate the risk of RT.

Notably, emerging trials such as STELLAR now stratify patients according to treatment history, distinguishing between treatment-naïve and previously treated populations, in recognition of their differing prognosis [36]. This trial is scheduled for completion in 2027 and reflects a broader shift toward personalized and risk-adapted therapeutic strategies in RT.

MRD is a validated prognostic tool in CLL, typically measured by multicolor flow cytometry (~10^−4^ sensitivity) or next-generation sequencing (~10^−6^) [37,38]. Its prognostic role in RT remains undefined and should be interpreted cautiously. Recent CAR T-cell therapy studies in RT report high response rates but have not consistently integrated MRD assessment into outcome measures CLL [39,40]. While MRD negativity pre-allo-HSCT correlates with superior outcomes in aggressive lymphoma cohorts, specific data in RT are lacking [40,41]. Prospective RT trials incorporating MRD as a biomarker to guide consolidation strategies would be worthwhile.

## 4. Treatment Outcomes Across CIT, BTKi, and BCL2 Inhibitor Eras

Multiple treatment strategies are now available for managing RT (Table 2). In this review, response definitions are based on the Lugano 2014 criteria for lymphoma. Complete response (CR) is characterized by Positron Emission Tomography/Computed Tomography (PET/CT) negativity and complete resolution of disease. Partial response is defined as a reduction of at least 50% in measurable disease without evidence of new lesions. Relapse denotes the presence of new or progressive diseases. MRD refers to subclinical disease identified through flow cytometry or molecular methods [42]. Historically, RT was treated as de novo DLBCL using conventional CIT regimens such as R-CHOP [42]. However, outcomes have been poor, with CR rates of only around 20% and median OS ranging from 6 to 12 months [6,43]. Attempts to improve efficacy through intensified regimens, such as hyper-CVAD or platinum-based combinations, have yielded only modest improvements in response rates without meaningful extension of OS [27,44,45,46,47]. For example, dose-adjusted R-EPOCH has demonstrated response rates as high as 67% in some series; however, its benefit is limited by early relapse, particularly among patients with complex karyotypes [48]. The French Richter study similarly reported an overall response rate (ORR) of 60% using platinum-based CIT, yet 2-year survival without transplant remained under 20% [27,44].

BTKi used as monotherapy in established RT have shown limited efficacy. Ibrutinib and acalabrutinib yield ORRs of 38 to 40% with median progression-free survival (PFS) of 3 to 4 months and few durable responses [49,50,51]. Zanubrutinib plus the PD-1 inhibitor tislelizumab modestly improved ORR to 58% in a phase II study [52]. Non-covalent BTKi such as pirtobrutinib have also demonstrated encouraging activity, with an ORR of 50% and a CR rate of 13%; however, the median PFS remained limited at 3.7 months [53]. These findings suggest that while BTKi may provide partial disease control in RT, they are rarely sufficient as monotherapy.

Venetoclax, a BCL2 inhibitor, has shown promise in RT, especially in combination. Although RT can develop during venetoclax treatment, often early in the clinical course, responses to venetoclax monotherapy are minimal, with a reported median OS of only 1.1 months in affected patients [54,55]. In contrast, combining venetoclax with CIT has yielded more. In a phase I study, the combination of venetoclax with dose-adjusted R-EPOCH (VR-EPOCH) achieved an ORR of 90% and a CR rate of 53% [56]. A multicenter phase II trial evaluating venetoclax plus R-CHOP (VR-CHOP) reported an ORR of 65% and a CR rate of 46% [57]. Retrospective analyses further support the efficacy of venetoclax-based combinations, particularly when paired with CIT, with ORR up to 55% and some patients achieving durable remissions following allo-HSCT [58]. The MOLTO trial investigated a chemotherapy-free regimen combining venetoclax, obinutuzumab, and the PD-L1 inhibitor atezolizumab, which resulted in an ORR of 67.9% and a CR rate of 28.6%, demonstrating that deep and sustained responses are achievable without cytotoxic chemotherapy [59,60]. Additional ongoing studies are evaluating venetoclax in combination with duvelisib or as part of triplet regimens with BTKi and anti-CD20 antibodies, aiming to further improve treatment outcomes in RT [61,62]. Despite encouraging results from combination regimens, BTKi monotherapy in RT generally yields only modest ORR and median PFS of 3 to 4 months, with few durable remissions [49,50,51]. Venetoclax monotherapy has shown limited efficacy, with some series reporting median OS as short as 1.1 months in affected patients [54,55]. It is important to note that most studies evaluating CIT, BTKi, and BCL2 inhibitor–based regimens in RT are retrospective or early-phase single-arm trials. These designs are inherently prone to selection bias, heterogeneous patient populations, and non-standardized response assessments, which limit the ability to directly compare outcomes across studies or to generalize results to all patients with RT.

Altogether, while conventional CIT remains inadequate, venetoclax-based combinations, particularly when used as a bridge to stem cell transplantation or cellular therapies, represent a significant advancement in the treatment of RT.

## 5. Role of Allo-HSCT

Allo-HSCT remains the only potentially curative treatment option for a subset of patients with RT [63], although it is associated with both significant benefits and notable limitations. Given the limited durability of response to CIT and novel agents, allo-HSCT is typically considered as a consolidative strategy for eligible patients who achieve remission after induction therapy [64,65,66]. In practice, allo-HSCT eligibility for RT typically requires Eastern Cooperative Oncology Group performance status 0–2, preserved organ function, and controlled infection. Conditioning regimens are often adjusted based on age, with reduced-intensity protocols offered to older patients to mitigate toxicity. Pre-transplant comorbidities are commonly assessed using the Hematopoietic Cell Transplantation–Specific Comorbidity Index, which has been shown to predict treatment-related mortality (TRM) and OS [67,68,69,70]. These patient- and disease-related factors, along with disease status at transplant, donor availability, and patient preferences, explain why only a subset of RT patients proceed to allo-HSCT.

The depth of response prior to allo-HSCT is a critical prognostic factor in RT. Prior study has shown that patient who undergo transplant in a responsive disease status have better outcomes than those transplanted with active disease [27]. The essential pre-transplant response parameters are achieving a radiographic CR on PET/CT (no sites of active uptake) and clearance of disease on biopsy (bone marrow and any previously involved nodes)—essentially, a CR by Lugano 2014 criteria. Patients in CR have significantly better transplant outcomes [71]. Partial responders can still proceed to transplant if that is the best response achievable, but their post-transplant outcomes are less optimal than those of patients in CR. In contrast, neither guidelines nor studies currently require molecular remission that achieved negative MRD before transplant, as its prognostic significance in RT remains undefined. The priority is achieving a clinical and metabolic remission, which has demonstrable impact on improving post-transplant survival in patients with RT.

Although no randomized trials have been conducted, multiple retrospective studies have demonstrated that long-term survival is possible after transplant in RT. In the Mayo Clinic series, 24 of 204 RT patients underwent transplantation (20 autologous, 4 allogeneic); the median OS post-transplant was 55.4 months, significantly longer than the approximately 12-month median OS in non-transplanted patients [6]. However, these results should be interpreted with caution, as transplant recipients were highly selected individuals who were younger, had fewer comorbidities, and were in remission at the time of transplant.

The Center for International Blood and Marrow Transplant Research reported outcomes from a multi-center cohort, showing a 3-year OS of 43 to 61% and PFS of 34 to 52% after allo-HSCT [71]. In contrast, autologous-HSCT offered slightly better short-term outcomes due to lower TRM. However, the tradeoff was a higher relapse rate and limited long-term survival. The curative potential of allo-HSCT is largely attributed to the graft-versus-lymphoma effect, although it carries significant high risks. TRM remains substantial, with early transplant-related deaths approaching 25% in the study [72]. In a retrospective study conducted by the European Group for Blood and Marrow Transplantation, among 59 patients with RT, approximately half received allo-HSCT. The 2-year OS was 39% for allogeneic and 36% for autologous HSCT [41,73]. Notably, only patients receiving allogeneic grafts achieved sustained remissions beyond 3 years, further supporting allo-HSCT as the preferred curative strategy. These findings suggest that although autologous-HSCT may be considered for patients with chemosensitive disease who are not eligible for allo-HSCT, it is less likely to achieve durable remission due to the persistent risk of relapses from residual CLL or transformed clones. Therefore, autologous-HSCT should not be regarded as a first-line curative option.

Despite these promising data, many patients with RT are older and have comorbidities due to prior CLL treatment, limiting their eligibility for myeloablative conditioning. Reduced-intensity conditioning (RIC) allo-HSCT regimens have been explored as an alternative. For example, an MD Anderson series using RIC in 8 patients with RT reported that 3 remained alive and disease-free at over 3 years [73,74]. A French multicenter study of 17 patients reported a 2-year OS of 47%, though relapse and non-relapse mortality remained high [75].

An “intention-to-treat” analysis from Canada highlighted the challenge of reaching transplant: many patients referred never underwent transplantation due to rapid disease progression or fitness decline [72]. This underscores the need for early referral and effective induction strategies to enable timely transplant.

Allo-HSCT remains the only potentially curative therapy for RT; however, relapses following transplant is a concern, prompting interest in post-transplant maintenance strategies. Given the rarity and heterogeneity of RT, maintenance therapy after allo-HSCT remains an area with very limited evidence. Donor lymphocyte infusion for early relapses is the one strategy with some documented success [76,77]. A recent multicenter study by Eikema et al. briefly discussed donor lymphocyte infusion and maintenance therapies in high-risk RT patients undergoing allo-HSCT, although no standardized maintenance regimen was uniformly employed [41]. Targeted agents, including venetoclax, ibrutinib, and checkpoint inhibitors, have not been validated as post-transplant maintenance therapies, and their routine use is not supported in the absence of trial data.

However, allo-HSCT is associated with substantial early and late toxicities, including acute and chronic graft-versus-host disease, organ toxicities, prolonged cytopenias, and treatment-related mortality rates up to 20 to 25%. Long-term survivors often experience moderate to severe late complications affecting multiple organ systems, with fatigue and insomnia remaining common [78].

Overall, allo-HSCT provides the best chance of long-term remission or cure in RT, especially for younger, fit patients who achieve CR. Early transplant consultation should be considered in all eligible patients with RT responding to initial therapy. Autologous transplantation may postpone relapses in certain cases, though relapses occur frequently and long-term remissions are uncommon. The available allo-HSCT data for RT are derived almost entirely from retrospective series with small sample sizes and significant selection bias toward younger, fitter patients who achieve remission before transplant. The absence of prospective randomized studies means that reported survival benefits must be interpreted with caution, and outcomes may not be replicable in broader clinical practice. Future directions include improving induction regimens to enable more patients to reach transplant, exploring maintenance strategies (e.g., post-alloHSCT BTKi or BCL2 inhibitor), and evaluating CAR T-cell therapy as a consolidative or alternative approach.

## 6. CAR T-Cell Therapy in RT

Anti-CD19 CAR T-cell therapy has revolutionized the management of relapsed or refractory DLBCL (r/r DLBCL). Given that RT often presents as an aggressive, CD19-positive lymphoma, there has been substantial interest in applying CAR T-cell therapy to this setting. However, the application of CAR-T cell therapy in patients with RT involves several practical considerations (Table 3).

In the previous pivotal clinical trial of anti-CD19 CAR T-cell therapy for r/r DLBCL, patients with RT were not included because RT is regarded as a distinct disease entity [79,80]. While no CAR T-cell product is specifically approved for RT, several retrospective studies and clinical trials have evaluated its efficacy, largely through off-label use (Table 4).

In CLL, the CR rate for lisocabtagene maraleucel (liso-cel) was 18%, which is lower than the rate observed in DLBCL, and infusion products often show high levels of T-cell exhaustion [87,88,89]. This raises the concern that CAR T-cell therapy may have reduced efficacy in RT compared to DLBCL. RT differs immunologically from CLL, showing high PD1 expression and strong responses to PD1 inhibitors [52,90,91], while CLL shows minimal response to PD1 inhibition [92].

An international multicenter retrospective study assessed 69 patients with RT treated with commercial anti-CD19 CAR T-cell therapies—axicabtagene ciloleucel, tisagenlecleucel, or liso-cel—following extensive prior therapy, including BTKi and BCL-2 inhibitors in 84% of cases [82]. The ORR was 63%, with a CR rate of 46%. Those who achieved CR experienced durable responses, with a median duration of response (DoR) of 27.6 months. Of note, detectable CLL MRD was eradicated in 81% of patients. A 2025 multicenter analysis confirmed durable remissions in a subset of patients, though relapse remains common, especially in TP53-aberrant cases [93].

Two additional retrospective studies have corroborated that CAR T-cell therapy for RT yields response rates comparable to those observed in de novo DLBCL. One study compared outcomes between patients with RT (*n* = 30) and those with aggressive B-cell lymphoma (*n* = 283) undergoing CAR T-cell therapy, finding similar safety profiles but reduced efficacy in the RT group [81]. At day 100, the ORR and CR in RT patients were 57% and 47%, respectively. Furthermore, RT patients exhibited inferior median (9.9 vs. 18 months) and 12-month OS rates (45% vs. 62%) compared to DLBCL, with worse survival associated with higher numbers of prior therapies, elevated LDH, and RT histology. In another international retrospective analysis conducted by the European research initiative on CLL, an ORR of 65% and a CR rate of 50% were reported among 54 RT patients receiving CAR T-cell therapy [83].

CAR T-cell therapy is associated with notable acute toxicities, including cytokine release syndrome (CRS) and immune effector cell-associated neurotoxicity syndrome (ICANS), which may present with confusion, language disturbances, and cognitive deficits [94]. Notably, even in a predominantly older and heavily pretreated population, toxicity rates were manageable, with grade ≥3 CRS and ICANS occurring at frequencies comparable to those observed in patients with transformed indolent non-Hodgkin lymphoma and de novo DLBCL [81]. Long-term follow-up data indicate that approximately 38 to 50% of survivors experience cognitive difficulties, anxiety, or depression [94,95,96]. Qualitative studies also report persistent impairments in domains such as sleep, role functioning, and emotional well-being, underscoring the importance of comprehensive survivorship care [97].

These initial response rates are similar to those seen in pivotal trials for r/r de novo DLBCL [79,80,98], indicating that RT cells remain sensitive to CAR T-cell cytotoxicity. However, response durability is lower; two retrospective studies with 2-year median follow-up reported median PFS of 4.7 and 4.3 months, and median OS of 8.5 and 9.9 months, respectively [81,82]. These outcomes are notably poorer than those observed in prior pivotal trials of anti-CD19 CAR T-cell therapy for de novo DLBCL [80,98,99].

The reasons for reduced durability in RT remain under investigation. Potential contributing factors include a highly immunosuppressive CLL microenvironment that may impair CAR T-cell persistence, higher tumor burden at the time of infusion, and the expression of checkpoint ligands or loss of target antigen (CD19) in RT cells [40,81]. To address the limitations associated with response durability, post-CAR T-cell consolidation strategies—such as allo-HSCT—may improve long-term outcomes in selected patients, warranting further investigation. Additional approaches, including combination therapies, are currently under active exploration. Notably, two early-phase clinical trials have attracted attention. One trial is evaluating the concurrent administration of a PD-1 checkpoint inhibitor (nivolumab) and a BTKi (ibrutinib) around the time of CAR T-cell infusion, based on the premise that ibrutinib may enhance T-cell fitness while nivolumab mitigates PD-1–mediated suppression [90]. Another trial is investigating CAR T-cell therapy in combination with zanubrutinib, a next-generation BTKi, in cases of relapsed RT (NCT05873712). These studies aim to overcome tumor microenvironment barriers and promote sustained CAR T-cell persistence.

Furthermore, next-generation therapeutic strategies are in development, including CAR T-cell constructs targeting alternative antigens in response to CD19 loss observed in certain RT cases, as well as modified cell therapies engineered to resist exhaustion or evade immunosuppressive mechanisms.

In summary, CD19 CAR T-cell therapy offers a promising but still limited option for patients with RT. The response rates are comparable to those in de novo DLBCL, but the majority of RT patients relapse within the first year, resulting in modest improvement of median PFS and OS. Nevertheless, CAR T-cell therapy remains an important treatment avenue for patients who are ineligible for allo-HSCT and have exhausted chemotherapy and novel agents. It may also serve as a bridge to transplant or long-term remission in a subset of patients. While CAR T-cell therapy achieves high initial responses, two large retrospective series reported median PFS of only 4.3 to 4.7 months and median OS less than 10 months [81,82], highlighting the substantial risk of early relapse even after remission. Published CAR T-cell therapy data in RT primarily originates from retrospective, multicenter registries or small institutional experiences. These analyses often lack standardized eligibility criteria, uniform response assessment, and consistent follow-up durations, which may overestimate efficacy or underestimate late toxicities. Ongoing research into patient selection, optimal timing, and adjunctive therapies will be essential to fully realize the potential of CAR T-cell therapy in RT.

## 7. Bispecific Antibody Therapies

Bispecific T-cell engaging antibodies represent a novel and promising class of immunotherapy for B-cell malignancies, including RT. These agents function by simultaneously binding CD3 on T-cells and a B-cell antigen, most commonly CD20, thereby redirecting cytotoxic T-cell activity against malignant B-cells. Therefore, bispecific antibodies serve as a potential alternative, given that patients with CLL often exhibit T-cell dysfunction (Table 5) [100].

Epcoritamab, a subcutaneously administered CD3×CD20 bispecific IgG1 antibody, has demonstrated notable efficacy in LBCL and is now being evaluated in RT. In the ongoing EPCORE CLL-1 phase I/II trial, 35 evaluable RT patients received single-agent epcoritamab [103]. The ORR was approximately 50%, with 35% achieving CR, several of which were also MRD-negative. Median PFS was 12.8 months. Considering that these patients were heavily pretreated, these results are noteworthy. Cytokine release syndrome CRS, mostly grade 1–2, occurred but was manageable with step-up dosing. These results establish epcoritamab as a potent and potentially durable treatment for RT. Combination studies are ongoing to further enhance efficacy.

Mosunetuzumab, another CD3×CD20 bispecific antibody administered intravenously, is already approved for relapsed follicular lymphoma and has shown activity in DLBCL. In a first-in-human study, 20 RT patients received mosunetuzumab monotherapy [104]. Among them, the ORR was 40%, with 20% achieving CR. Some remissions were durable beyond 6 months. Toxicity was similar to other lymphomas, with low-grade CRS and minimal neurotoxicity. Though the CR rate was lower than with epcoritamab, these patients were heavily pretreated, and mosunetuzumab still showed meaningful single-agent activity. An ongoing trial (NCT06521996) is combining mosunetuzumab with CHOP chemotherapy in frontline RT, aiming to improve initial disease control and deepen remission.

Glofitamab, a CD3×CD20 bispecific with a unique 2:1 binding format, is approved in Europe for relapsed/refractory DLBCL. While dedicated RT-specific data are limited, case reports suggest clinical responses in RT patients, even inducing CR [100,101]. Ongoing expansion cohorts include RT patients, and the drug is considered a viable option, especially in patients who relapse after CAR T-cell therapy or are ineligible for it. CRS is manageable with standard step-up dosing and obinutuzumab pre-treatment.

Ongoing studies such as Euplagia-1 are testing dual-targeting CAR T and immune engagers with promising early activity [104]. Trials of blinatumomab, a CD19-directed BiTE, have reported encouraging results in RT, including durable MRD-negative responses in selected patients [105].

In summary, bispecific antibodies offer an important new therapeutic class in RT. Their key advantages include off-the-shelf availability, repeat dosing potential, and relatively manageable toxicity profiles. Epcoritamab in particular has shown promising response rates, including MRD-negative CR. As more data emerge, these agents may be positioned in multiple treatment lines—either as monotherapy in relapsed RT, in combination regimens, or following CAR T-cell failure. Future trials will determine whether these therapies can serve as curative platforms, particularly when paired with consolidation strategies. Bispecific antibodies are poised to become a cornerstone in the evolving management of RT.

## 8. Ongoing and Future Clinical Trials

The urgent need for more effective therapies in RT has spurred a wave of clinical trials exploring novel strategies (Figure 2). These include combination regimens, immunotherapies, and next-generation cellular and targeted approaches in both frontline and relapsed settings. Below is a summary of key ongoing or planned trials and future directions.

### 8.1. Combination Chemo-Immunotherapy Trials

In addition to the STELLAR study, which compares standard R-CHOP with R-CHOP plus acalabrutinib [36], another ongoing trial (NCT04679012) is evaluating polatuzumab vedotin—an anti-CD79b antibody-drug conjugate—in combination with R-CHOP. Since CD79b is frequently expressed in RT cells, the addition of polatuzumab may potentiate R-CHOP efficacy. Preliminary results will guide further development of this chemo-immunotherapy strategy.

### 8.2. Checkpoint Inhibitor Combinations

Recent data suggest a potential role for checkpoint blockade in RT. A phase II study of pembrolizumab demonstrated responses in heavily pretreated RT patients, particularly those with prior BTKi exposure [106]. Copanlisib plus nivolumab also showed activity, with manageable toxicity, in a multicenter trial [107]. Ongoing work also includes duvelisib with nivolumab (NCT03892044), based on the hypothesis that PI3K inhibition may modulate the tumor microenvironment to enhance T-cell responses. Early data from a phase II trial also suggest that combining pembrolizumab with BTKi may improve outcomes compared to PD-1 monotherapy [108].

### 8.3. Triple Targeted Therapy Regimens

“Time-limited” triplet therapies are gaining traction. Trial NCT05536349 investigates pirtobrutinib (non-covalent BTKi), venetoclax, and obinutuzumab in newly diagnosed CLL and RT. This trial’s inclusion of RT in a frontline CLL cohort is novel and may illuminate strategies to intercept early transformation. Another trial, DTRM-555, evaluates a triplet of investigational BTKi DTRMWXHS-12, everolimus (mTOR inhibitor), and pomalidomide (an immunomodulator) [109]. Preliminary findings indicate certain responses, but myelosuppression remains a potential issue. Not all investigational regimens have been successful. The phase II study was discontinued in December 2024 due to financial constraints [110]. This underscores the challenges of developing effective therapies in a rare and heterogeneous disease. These studies test whether targeting multiple oncogenic pathways simultaneously can suppress RT escape mechanisms.

### 8.4. CAR T-Cell Therapy Enhancements

New strategies seek to improve CAR T-cell efficacy in RT. One NCI-sponsored trial (NCI-2022-10247) combines nivolumab and ibrutinib with CD19 CAR T-cells to enhance expansion and persistence. Another study (NCI-2023-03669) pairs zanubrutinib with CD19 CAR T therapy. Exploratory platforms using “armored” CAR T-cells, which are engineered to express cytokines or PD-1 dominant negative receptors, are being developed specifically for RT. International consortium efforts are also collecting real-world data to refine CAR T-cell use in this population [111].

### 8.5. Bispecific Antibodies and Emerging Immunotherapies

Building on early success, bispecific antibodies are moving into combination trials. Epcoritamab is being evaluated in the ongoing EPCORE CLL-1 trial, with expansion cohorts testing it in RT and planned combinations with BTKi or venetoclax (NCT04623541). Mosunetuzumab is being combined with CHOP in the frontline RT setting [111]. Additionally, immunotherapies targeting macrophage checkpoints like CD47 are being studied. Evorpacept (ALX148), a CD47-blocking agent, showed early signs of activity in aggressive lymphomas and may be explored in RT [112]. Novel immune cell engagers, such as NK-cell or next-gen T-cell engagers targeting CD19, ROR1, and others, are under preclinical evaluation [113,114,115].

### 8.6. Targeting ROR1 and Novel Molecular Pathways

ROR1, an embryonic surface protein, is highly expressed in CLL and may persist in RT. A novel antibody-drug conjugate, zilovertamab vedotin, targets ROR1 and has shown activity in mantle cell lymphoma and CLL [116]. The planned waveLINE-006 phase 2 trial will test this agent in aggressive B-cell lymphomas, potentially including RT. Other pathways under investigation include MYC, BCL6, and CDK9. For instance, selective CDK9 inhibitors are being tested to overcome venetoclax resistance by downregulating MCL1 [115]. Molecular profiling of RT biopsies continues to search for targetable mutations unique to RT, though most mutations currently overlap with high-risk CLL.

## 9. Conclusions

RT remains highly aggressive and treatment-resistant, with limited long-term survival despite recent therapeutic advances. Conventional CIT offers poor durability, while allogeneic stem cell transplantation may provide a potential cure but carries substantial non-relapse mortality, particularly in older or frail patients. Newer approaches have shown promise in select cases, including BTKi and BCL2 inhibitors, CAR T-cell therapy, and bispecific antibodies. Bispecific antibodies like epcoritamab are especially encouraging as off-the-shelf immunotherapies. Ongoing clinical trials are exploring combination strategies and novel targets such as ROR1 and CD47. Continued research and early referral remain critical to improving outcomes in this high-risk population.

## Figures and Tables

**Figure 1 ijms-26-08747-f001:**
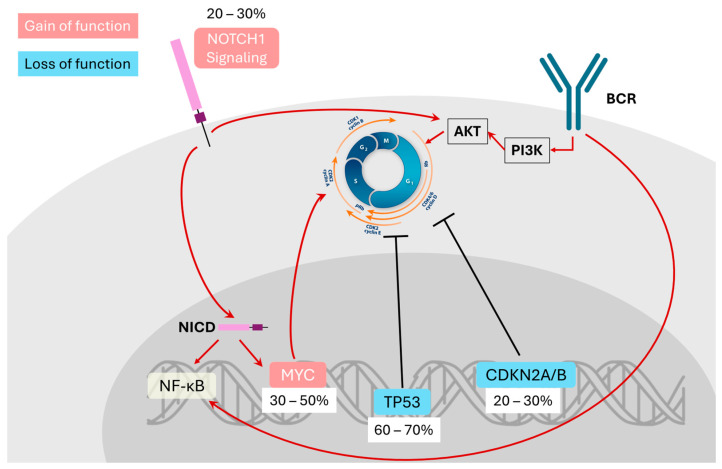
Molecular mechanisms underlying Richter transformation. This diagram illustrates how chronic B-cell receptor (BCR) activation interacts with recurrent genetic lesions to drive the pathogenesis of Richter transformation (RT). BCR signaling—often delivered through unmutated or stereotyped IGHV receptors (e.g., subset #8)—activates downstream pathways such as PI3K–AKT and NF-κB. Loss of tumor-suppressor checkpoints (*TP53* mutation/deletion and *CDKN2A/B* loss) removes cell-cycle brakes and apoptotic safeguards, enabling BCR-dependent proliferation of large pleomorphic cells. *NOTCH1* mutations, which are associated with biased subset-8 BCR usage and autonomous signaling, potentiate PI3K/AKT and NF-κB pathway activation and markedly increase the risk of transformation. These pathways converge on *MYC*, promoting metabolic reprogramming and rapid proliferation. Constitutive AKT phosphorylation—observed in high-risk CLL with *NOTCH1* or *TP53* alterations—further underscores the interplay between BCR, PI3K/AKT and *NOTCH1* signaling. Together, chronic antigenic drive and cumulative genetic lesions produce profound genomic instability (chromothripsis, chromoplexy, and whole-genome doubling), resulting in the emergence of aggressive RT clones.

**Figure 2 ijms-26-08747-f002:**
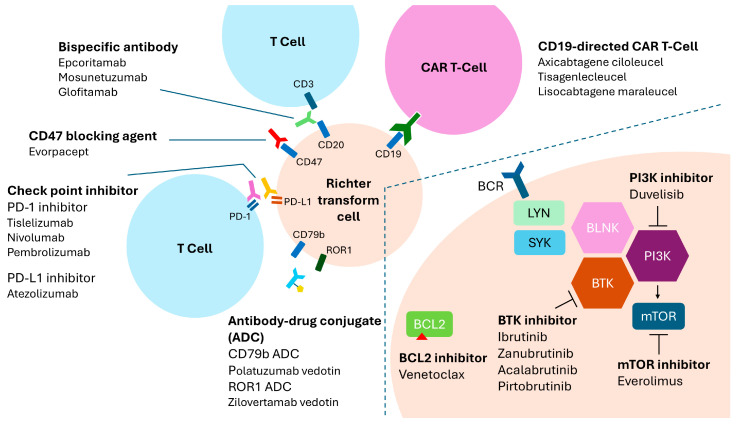
Targeted therapies under investigation for Richter transformation. The diagram highlights key therapeutic targets on transformed B cells and associated immune or signaling interactions. Shown are CD19-directed CAR T-cell products, CD20×CD3 bispecific antibodies, checkpoint inhibitors (PD-1/PD-L1), CD47 blockade, antibody–drug conjugates (ADCs), and small-molecule inhibitors targeting BCL2, BTK, PI3K, and mTOR. These agents represent mechanistically distinct approaches being combined to improve outcomes in Richter transformation.

**Table 1 ijms-26-08747-t001:** Incidence of Richter Transformation in Different Studies.

Study (Year)	Setting	RT Incidence
MDACC (1975 to 2005) [27]	CIT era, *N* = 3986 Retrospective	5.1%
GCLLSG (1999 to 2016) [28]	CIT era, *N* = 2975 Pooled analysis of trials	2–10%
Nationwide Danish (2008 to 2016) [19]	Retrospective *N* = 3772	2.6% DLBCL-RT
SEER registry [29]	Retrospective *N* = 74,166	0.7%
RESONATE [30] (2012 to 2018 follow-up)	Phase 3, RCT, r/r CLL Ibrutinib, *n* = 195	5.1%
RESONATE-2 [31] (2013 to 2018 follow-up)	Phase 3, RCT, Ibrutinib, *n* = 136	1.5%
MURANO [32] (2014 to 2015)	Phase 3, RCT, r/r CLL Venetoclax + Rituximab, *n* = 194 Bendamustine + Rituximab, *n* = 195	V + R, 3.1% B + R, 2.6%

Abbreviations: CIT, chemoimmunotherapy; R/R, relapsed/refractory; RT, Richter transformation.

**Table 2 ijms-26-08747-t002:** Comparison of Therapeutic Strategies in RT.

Therapy	Pros	Cons	Comments
Traditional standard therapies			
Chemoimmunotherapy	Broad clinical accessibility Induces rapid cytoreduction Established use in standard practice	Low CR rate (~20–30%) Median OS of 6–12 months High frequency of relapse	Historically used as front-line therapy Suboptimal for durable disease control Often serves as induction prior to other approaches
Allo-HSCT	Only potentially curative approach Enables long-term remissions (>3 years) Leverages graft-versus-lymphoma effect	High TRM (up to 25%) Restricted to transplant-eligible patients Requires disease CR prior to allo-HSCT	Optimal for younger, fit patients in remission Early referrals Reduced-intensity conditioning expand eligibility
Novel approaches			
BTK Inhibitors	Oral administration Active in a subset of RT cases Favorable tolerability	Modest ORR (~38–58%) Short PFS (3–4 months) Common resistance and early relapses	Frequently used as salvage or bridging therapy Combination strategies may enhance efficacy Limited clinical benefit as monotherapy
BCL2 Inhibitors	Encouraging in combination regimens ORR up to 90% Chemotherapy-free options	Limited efficacy as monotherapy High relapse rates with single-agent use Potential risk of tumor lysis syndrome	Most effective in combination regimens Commonly utilized as a bridge to allo-HSCT or CAR T-cell therapy
CAR T-cell Therapy	High ORR (63–65%) Achieve MRD-negative CR in some cases Demonstrated efficacy in off-label use	Short DOR Limited accessibility T-cell dysfunction and exhaustion Risk of immune-related toxicities	Promising option for chemo-refractory RT Effective as a bridge to transplant in responders Potential for enhanced efficacy in combination strategies
Bispecific Antibodies	Off-the-shelf availability Repeat dosing for sustained control ORR of 40–50% reported in RT Manageable toxic profile	Long-term DOR remains uncertain Limited data available Optimal sequencing with CAR T-cell therapy or allo-HSCT is not yet defined	An emerging option, especially for post-CAR T relapses or patients ineligible for cellular therapy Subcutaneous administration enhances accessibility and treatment adherence

Abbreviations: Allo-HSCT, allogeneic hematopoietic stem cell transplantation; BCL2, B-cell lymphoma 2; BTK, Bruton’s tyrosine kinase; CAR T, chimeric antigen receptor T cell; CR, complete response; MRD, minimal residual disease; ORR, overall response rate; OS, overall survival; PFS, progression-free survival; RT, Richter transformation.

**Table 3 ijms-26-08747-t003:** Key Considerations and Clinical Challenges of CAR T-cell Therapy in Patients with RT.

Considerations	Comments
Aggressive Disease	RT often harbors *TP53* mutations, complex karyotype, and unmutated IGHV, all of which predict poor outcomes with conventional therapy.
Manufacturing Barriers	Rapidly progressing RT necessitates bridging therapy, which may impair CAR T-cell fitness. Lymphodepleting regimens before infusion must be cautiously managed to avoid toxicity.
Toxicity	CAR T-cell related adverse events like CRS and ICANS are prominent and more severe in RT patients due to high tumor burden. Infection risk is elevated due to heavy pretreatment and aggressive disease.
On-Target, Off-Tumor Toxicity and CD19 Escape	Loss of CD19 antigen expression can lead to CAR T-cell resistance. RT cells may show heterogeneous CD19 expression, complicating targeting.
Limited Data	RT patients are often excluded from most pivotal CAR T-cell trials.
Patient Factors	RT patients are often older and frailer, with prior toxicities from CLL therapies, limiting eligibility for CAR-T. Functional status and disease kinetics must guide decision-making.

Abbreviations: CAR-T, chimeric antigen receptor T cell; CLL, chronic lymphocytic leukemia; CRS, cytokine release syndrome; ICANS, immune effector cell-associated neurotoxicity syndrome; IGHV, immunoglobulin heavy chain variable region; RT, Richter transformation; *TP53*, tumor protein p53.

**Table 4 ijms-26-08747-t004:** Reported Outcomes of Anti-CD19 CAR T-Cell Therapies in Richter Transformation.

Reference	Patient Number	Prior Treatment Lines (Median)	ORR/CR (%)	PFS/OS (mo)
Benjamini O et al. [81]	30 (Axi-cel, *n* = 4; Tisa-cel, *n* = 7; Liso-cel, *n* = 3, POC anti-CD19 CAR T, *n* = 16)	2 (0–7)	57/40	4.3/9.9
Kittai AS et al. [82]	69 (Axi-cel, *n* = 44; Tisa-cel, *n* = 17, Liso-cel, *n* = 7; Brexu-cel, *n* = 1)	2 (0–7)	63/46	4.7/8.5
Beyar-Katz, O. et al. [83]	54 (Axi-cel, *n* = 4; Tisa-cel, *n* = 20; Liso-cel, *n* = 1, POC anti-CD19 CAR T, *n* = 29)	2 (0–8)	65/46	12-m PFS:41%/14.4
Kittai AS et al. [84]	Axi-cel, *n* = 8	4	100/63	NR/NR
Abramson, J.S. et al. [85,86]	Liso-cel, *n* = 4	NR	50/25	NR/NR

Axi-cel: Axicabtagene ciloleucel; Brexu-cel: Brexucabtagene autoleucel; CAR: Chimeric antigen receptor; CR: Complete response; Liso-cel: lisocabtagene maraleucel; mo: month; NR: Not reported; ORR: Overall response rate; OS: Overall survival; PFS: Progression-free survival; POC: Point-of-care; Tisa-cel: Tisagenlecleucel.

**Table 5 ijms-26-08747-t005:** Comparison of Bispecific Antibody Therapies in RT.

Bispecific Antibody	Trial Phase	Patient Numbers	ORR/CR (%)	PFS, Median (mo)
Epcoritamab [101]	I/II	35	50/35	12.8
Mosuntuzumab [102]	I/II	20	40/20	NR

Both studies do not report overall survival. Abbreviations: CR, complete response; mo, months; NR, not reported; ORR, overall response rate; PFS, progression-free survival.

## Data Availability

Not applicable.

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
