# Peer review of "Richter Transformation in Chronic Lymphocytic Leukemia: Current Treatment Challenges and Evolving Therapies"

_ijms, 2025, doi:10.3390/ijms26178747_

Round 1
Reviewer 1 Report
Comments and Suggestions for Authors
The paper offers a detailed landscape of treatment options for Richter Transformation (RT), including traditional chemoimmunotherapy (CIT), targeted agents like BTK inhibitors, BCL2 inhibitors, cellular therapies like CAR T-cells, and allogeneic hematopoietic stem cell transplantation (allo-HSCT).
Overall, the review provides a solid, data-driven overview of the evolving therapeutic landscape for RT. Its comprehensive approach and inclusion of recent advances are commendable. To enhance its impact, a more nuanced critique of the evidence base, consideration of patient-centered outcomes, and explicit discussion of research gaps would be beneficial. This would better guide clinicians in translating these findings into practice and identifying areas for future research.
Major points
- While the paper summarizes current data, it offers limited critique of the quality of evidence. For example, most data are from retrospective studies or early-phase trials; the paper could better emphasize the limitations inherent in these study designs, such as bias, small sample sizes, or lack of control groups.
- The criteria for defining remission, response, or relapse (e.g., "deep remission," "partial response") are mentioned but not standardized or contextualized. Clarification on how these definitions influence outcomes or decision-making would improve clinical applicability.
- While efficacy is extensively discussed, there is minimal focus on the toxicity profiles, long-term side effects, or patient-reported outcomes associated with these therapies, especially with intensive procedures like allo-HSCT or CAR T-cell therapies.
- The paper predominantly cites positive results or promising data from newer treatments. It could benefit from a more balanced discussion of negative or neutral findings, or failed trials, to
- The review mentions that allo-HSCT is limited to transplant-eligible patients but does not delve into the criteria for eligibility or how patient comorbidities impact outcomes, which is crucial for real-world application.
Author Response
Reviewer 1
The paper offers a detailed landscape of treatment options for Richter Transformation (RT), including traditional chemoimmunotherapy (CIT), targeted agents like BTK inhibitors, BCL2 inhibitors, cellular therapies like CAR T-cells, and allogeneic hematopoietic stem cell transplantation (allo-HSCT).
Overall, the review provides a solid, data-driven overview of the evolving therapeutic landscape for RT. Its comprehensive approach and inclusion of recent advances are commendable. To enhance its impact, a more nuanced critique of the evidence base, consideration of patient-centered outcomes, and explicit discussion of research gaps would be beneficial. This would better guide clinicians in translating these findings into practice and identifying areas for future research.
Major points
Comments 1: While the paper summarizes current data, it offers limited critique of the quality of evidence. For example, most data are from retrospective studies or early-phase trials; the paper could better emphasize the limitations inherent in these study designs, such as bias, small sample sizes, or lack of control groups.
Response 1: Thank you for this valuable comment. We have revised the manuscript to explicitly acknowledge the limitations of the current evidence base for RT management. Specifically, we have added statements in Sections 4, 5, and 6 noting that much of the available data are derived from retrospective analyses, early-phase single-arm trials, and small patient cohorts, with inherent risks of selection bias, heterogeneous populations, and non-standardized response assessments. We have also emphasized that the absence of randomized controlled trials limits the generalizability and comparability of reported outcomes. These additions provide a more critical appraisal of the evidence and better contextualize the reported results.
In section “4. Treatment Outcomes Across CIT, BTKi, and BCL2 Inhibitor Eras”, line 194 to 205: Despite encouraging results from combination regimens, BTKi monotherapy in RT generally yields only modest overall response rates and median PFS of 3–4 months, with few durable remissions [45-47]. Venetoclax monotherapy has shown limited efficacy, with some series reporting median OS as short as 1.1 months in affected patients [50, 51]. It is important to note that most studies evaluating CIT, BTKi, and BCL2 inhibitor–based regimens in RT are retrospective or early-phase single-arm trials. These designs are inherently prone to selection bias, heterogeneous patient populations, and non-standardized response assessments, which limit the ability to directly compare outcomes across studies or to generalize results to all patients with RT.”
In section “5. Role of Allo-HSCT”, line 287 to 292: “The available allo-HSCT data for RT are derived almost entirely from retrospective series with small sample sizes and significant selection bias toward younger, fitter patients who achieve remission before transplant. The absence of prospective randomized studies means that reported survival benefits must be interpreted with caution, and outcomes may not be replicable in broader clinical practice.”
In section “6. CAR T-cell Therapy in RT”, line 388 to 391: “Published CAR T-cell therapy data in RT primarily originate from retrospective, multicenter registries or small institutional experiences. These analyses often lack standardized eligibility criteria, uniform response assessment, and consistent follow-up durations, which may overestimate efficacy or underestimate late toxicities.”
Comments 2: The criteria for defining remission, response, or relapse (e.g., "deep remission," "partial response") are mentioned but not standardized or contextualized. Clarification on how these definitions influence outcomes or decision-making would improve clinical applicability.
Response 2: Thank you for this valuable suggestion. Section 4 now define the response criteria from line 150 to 156: “In this review, response definitions are based on the Lugano 2014 criteria for lymphoma. Complete remission (CR) is characterized by PET negativity and complete resolution of disease. Partial response is defined as a reduction of at least 50% in measurable disease without evidence of new lesions. Relapse denotes the presence of new or progressive diseases. Minimal residual disease (MRD) refers to subclinical disease identified through flow cytometry or molecular methods” This addition standardizes and contextualizes the definitions of remission, response, and relapse, thereby improving the clinical applicability of our discussion.
Comments 3: While efficacy is extensively discussed, there is minimal focus on the toxicity profiles, long-term side effects, or patient-reported outcomes associated with these therapies, especially with intensive procedures like allo-HSCT or CAR T-cell therapies.
Response 3: We sincerely appreciate your insightful comments. In response, we have revised the manuscript to incorporate a more detailed discussion of the toxicity profiles, long-term adverse effects, and – where data are available – patient reported outcomes (PROs) associated with intensive treatments such as allo-HSCT and CAR T-cell therapy.
The following sentences have been added to the revised manuscript:
Section 5, line 278 to 282: “However, allo-HSCT is associated with substantial early and late toxicities, including acute and chronic graft-versus-host disease, organ toxicities, prolonged cytopenias, and treatment-related mortality rates up to 20–25%. Long-term survivors often experience moderate to severe late complications affecting multiple organ systems, with fatigue and insomnia remaining common.”
Section 6, line 343 to 353: “CAR T-cell therapy is associated with notable acute toxicities, including cytokine release syndrome (CRS) and immune effector cell-associated neurotoxicity syndrome (ICANS), which may present with confusion, language disturbances, and cognitive deficits. Notably, even in a predominantly older and heavily pretreated population, toxicity rates were manageable, with grade ≥3 CRS and ICANS occurring at frequencies comparable to those observed in patients with transformed indolent non-Hodgkin lymphoma and de novo DLBCL. Long-term follow-up data indicate that approximately 38 to 50% of survivors experience cognitive difficulties, anxiety, or depression. Qualitative studies also report persistent impairments in domains such as sleep, role functioning, and emotional well-being, underscoring the importance of comprehensive survivorship care.”
Comments 4: The paper predominantly cites positive results or promising data from newer treatments. It could benefit from a more balanced discussion of negative or neutral findings, or failed trials, to
Response 4: Thank you for this insightful comment. We have revised the manuscript to include a more balanced discussion of newer treatments by incorporating negative or neutral findings and examples of unsuccessful or discontinued trials. These additions aim to provide readers with a realistic perspective on the limitations of current approaches.
Section 4 (BTKi and BCL2 inhibitors), line 194 to 202:
Despite encouraging results from combination regimens, BTKi monotherapy in RT generally yields only modest overall response rates and median progression-free sur-vival of 3–4 months, with few durable remissions. Venetoclax monotherapy has shown limited efficacy, with some series reporting median overall survival as short as 1.1 months in affected patients. It is important to note that most studies evaluating CIT, BTKi, and BCL2 inhibitor–based regimens in RT are retrospective or early-phase single-arm trials. These designs are inherently prone to selection bias, heterogeneous patient populations, and non-standardized response assessments, which limit the ability to directly compare outcomes across studies or to generalize results to all patients with RT.
Section 6 (CAR T-cell Therapy), line 385 to 388:“While CAR T-cell therapy achieves high initial responses, two large retrospective series reported median progression-free survival of only 4.3 to 4.7 months and median overall survival under 10 months, highlighting the substantial risk of early relapse even after remission.”
Section 8 (Ongoing and Future Clinical Trials), line 483 to 487:
“Preliminary findings indicate certain responses, but myelosuppression remains a potential issue. Not all investigational regimens have been successful. The phase II study was discontinued in December 2024 due to financial constraints. This underscores the challenges of developing effective therapies in a rare and heterogeneous disease”
Comments 5: The review mentions that allo-HSCT is limited to transplant-eligible patients but does not delve into the criteria for eligibility or how patient comorbidities impact outcomes, which is crucial for real-world application.
Response 5: Thank you for this important suggestion. We have revised Section 5 to include a concise description of commonly used eligibility criteria for allo-HSCT in RT and to highlight how comorbidities influence outcomes in real-world practice. This addition aims to provide readers with practical guidance on patient selection.
In section 5, line 211 to 219:
“In practice, allo-HSCT eligibility for RT typically requires ECOG performance status 0–2, preserved organ function, and controlled infection. Conditioning regimens are often adjusted based on age, with reduced-intensity protocols offered to older patients to mitigate toxicity. Pre-transplant comorbidities are commonly assessed using the Hematopoietic Cell Transplantation–Specific Comorbidity Index, which has been shown to predict treatment-related mortality (TRM) and overall survival. These patient- and disease-related factors, along with disease status at transplant, donor availability, and patient preferences, explain why only a subset of RT patients proceed to allo HSCT.”

Reviewer 2 Report
Comments and Suggestions for Authors
Summary:
In the presented literature review titled “Richter Transformation in Chronic Lymphocytic Leukemia: Current Treatment Challenges and Evolving Therapies”, Lin et al. summarize the biology, current and future treatments, and challenges in the management of Richter transformation (RT). The authors first summarize the pathologic and molecular features of RT, including its clonal origin, subtypes, major genomic features and overall prognosis. They then go on to describe the available treatments for RT and their respective prognosis, including BTK inhibitors, BCL2 inhibitors, allogeneic hematopoietic stem cell transplant (Allo-HSCT), bispecific antibodies and chimeric antigen receptor (CAR)-T cell therapy. The authors outline risks and benefits of these treatments, and ongoing/completed clinical trials. They further highlight future therapies under investigation. The article closes with a summary of remaining challenges for the treatment of RT, such as limited duration of treatment response, and the prevalent frailty/morbidity of RT patients.
General Concept comments:
The article is concisely written and clearly structured. It summarizes the evolving field of RT and RT treatments well and cites many recent studies in the field. It delivers clear summaries of treatment concepts for RT in several tables and one figure. The conclusions and discussion points highlighted by the authors are fully justified by the data/studies presented.
However, some aspects relevant to RT remain incompletely addressed. These include the role of minimal residual disease (MRD) and its detection modalities, the role of prognostic biomarkers and patient stratification in RT, and the risk factors predisposing to RT. While these aspects do not have to take center stage in the article, they should be mentioned and briefly described due to their importance in RT treatment planning. In section 3, the authors address the varying risk of RT in relation to different CLL frontline treatments. They highlight how RT incidence varies from 3% to up to 40%, depending on the study. Here, it would be beneficial to have a systematic summary of the incidence data and the studies they are derived from. This could for example be implemented in the form of a table. The authors may also discuss potential biological mechanisms, if known, of how different CLL treatments could lead to different RT incidence rates.
Overall, the article places strong focus on outlining different classes of RT treatments and presents these in a meaningful comparison. Additional aspects surrounding treatment decisions (MRD, risk factors, biomarkers) remain incompletely addressed, and should be mentioned and briefly explained in the manuscript.
Specific comments:
- Line 156: In table 2, the authors contrast risks and benefits of Allo-HSCT in RT. This point is somewhat redundant with table 1, where key points of Allo-HSCT are already mentioned. Table 2 also summarizes only one treatment and does so mostly in text form. This table should be combined with table 1, and relevant additional discussion points incorporated into the text body of section 5.
- Line 52, 116: The authors mention complex karyotype as a common factor associated with poor prognosis in RT. However, they do not specify how this term is defined. To improve the comprehensiveness of the article, the authors should briefly describe what a complex karyotype is, and how it has been defined.
- Line 171: The authors mention minimal residual disease (MRD), a key concept for the assessment of responses and potentially cure in CLL and RT. This aspect is not further followed up on. The authors should add some additional explanation about how MRD is assessed in RT, whether it has been successfully used in clinical trials, and where its future role in RT treatment might be. Some of that is implied in table 1 and section 6, but it should be presented more clearly in one place.
- Line 409: ”Novel immune cell engagers […] preclinical evaluation”. This statement should be backed up with a specific reference.
- Line 417: “selective CDK9 […] downregulating MCL1”. This statement should be backed up with a specific reference.
Author Response
Reviewer 2
Summary:
In the presented literature review titled “Richter Transformation in Chronic Lymphocytic Leukemia: Current Treatment Challenges and Evolving Therapies”, Lin et al. summarize the biology, current and future treatments, and challenges in the management of Richter transformation (RT). The authors first summarize the pathologic and molecular features of RT, including its clonal origin, subtypes, major genomic features and overall prognosis. They then go on to describe the available treatments for RT and their respective prognosis, including BTK inhibitors, BCL2 inhibitors, allogeneic hematopoietic stem cell transplant (Allo-HSCT), bispecific antibodies and chimeric antigen receptor (CAR)-T cell therapy. The authors outline risks and benefits of these treatments, and ongoing/completed clinical trials. They further highlight future therapies under investigation. The article closes with a summary of remaining challenges for the treatment of RT, such as limited duration of treatment response, and the prevalent frailty/morbidity of RT patients.
General Concept comments:
The article is concisely written and clearly structured. It summarizes the evolving field of RT and RT treatments well and cites many recent studies in the field. It delivers clear summaries of treatment concepts for RT in several tables and one figure. The conclusions and discussion points highlighted by the authors are fully justified by the data/studies presented.
However, some aspects relevant to RT remain incompletely addressed. These include the role of minimal residual disease (MRD) and its detection modalities, the role of prognostic biomarkers and patient stratification in RT, and the risk factors predisposing to RT. While these aspects do not have to take center stage in the article, they should be mentioned and briefly described due to their importance in RT treatment planning. In section 3, the authors address the varying risk of RT in relation to different CLL frontline treatments. They highlight how RT incidence varies from 3% to up to 40%, depending on the study. Here, it would be beneficial to have a systematic summary of the incidence data and the studies they are derived from. This could for example be implemented in the form of a table. The authors may also discuss potential biological mechanisms, if known, of how different CLL treatments could lead to different RT incidence rates.
Comments 1: Overall, the article places strong focus on outlining different classes of RT treatments and presents these in a meaningful comparison. Additional aspects surrounding treatment decisions (MRD, risk factors, biomarkers) remain incompletely addressed, and should be mentioned and briefly explained in the manuscript.
Response 1: We thank the reviewer for this valuable suggestion. We have revised the manuscript to briefly address additional clinically relevant aspects, including the role of minimal residual disease (MRD) and its detection modalities, prognostic biomarkers, and patient stratification in RT. This new subsection has been added at the start of section 3.
In Section 3, we now provide a systematic summary of reported RT incidence according to different CLL frontline treatments, presented in a new table with details on study type, incidence rate, and follow-up. We have also expanded the discussion to include potential biological mechanisms that may account for treatment-related differences in RT risk. These additions provide a more structured overview of incidence data and improve the applicability of our review to clinical decision-making.
The added text as follows:
Lines 102 to 105: High‑risk genomic alterations in CLL, including TP53 aberrations, NOTCH1 mutations, complex karyotype, and unmutated IGHV, are consistently associated with increased RT risk [1-8]. These factors, together with clinical features and prior therapy, support risk‑adapted treatment planning and trial referral in RT.
Lines 141 to 145: MRD is a validated prognostic tool in CLL, typically measured by multicolor flow cytometry (~10⁻⁴ sensitivity) or next-generation sequencing (~10⁻⁶ ) [46, 47]. Its prognostic role in RT remains undefined and should be interpreted cautiously. Recent CAR T-cell therapy studies in RT report high response rates but have not consistently integrated MRD assessment into outcome measures CLL [48, 49].
Reported RT incidence varies widely depending on treatment era, patient population, diagnostic criteria, and follow-up duration. A summary of major studies and cohorts is shown in Table 1.
|
Study (year) |
Setting |
RT incidence |
|
MDACC (1975 to 2005) [36] |
CIT era, N = 3986 Retrospective |
5.1% |
|
GCLLSG (1999 to 2016) [37] |
CIT era, N = 2975 Pooled analysis of trials |
2–10% |
|
Nationwide Danish (2008 to 2016) [19] |
Retrospective N = 3772 |
2.6% DLBCL-RT |
|
SEER registry [38] |
Retrospective N = 74,166 |
0.7% |
|
RESONATE [39] (2012 to 2018 follow-up) |
Phase 3, RCT, r/r CLL Ibrutinib, n = 195 |
5.1% |
|
RESONATE-2 [40] (2013 to 2018 follow-up) |
Phase 3, RCT, Ibrutinib, n = 136 |
1.5% |
|
MURANO [41] (2014 to 2015) |
Phase 3, RCT, r/r CLL Venetoclax+Rituximab, n = 194 Bendamustine+Rituximab, n = 195 |
V+R, 3.1% B+R, 2.6% |
Abbreviations: CIT, chemoimmunotherapy; R/R, relapsed/refractory; RT, Richter transformation.
Differences in incidence likely reflect patient selection, biopsy practices, treatment exposure, and biological effects of therapy. Hypothesized mechanisms include selection of pre-existing aggressive subclones under therapeutic pressure, therapy-induced immune dysfunction, and microenvironmental changes that facilitate transformation. For example, prolonged BTK inhibition may promote outgrowth of TP53-mutated clones, while venetoclax may preferentially spare apoptosis-resistant populations.
Specific comments:
- Line 156: In table 2, the authors contrast risks and benefits of Allo-HSCT in RT. This point is somewhat redundant with table 1, where key points of Allo-HSCT are already mentioned. Table 2 also summarizes only one treatment and does so mostly in text form. This table should be combined with table 1, and relevant additional discussion points incorporated into the text body of section 5.
Response: Thank you for your suggestion, we have deleted the table 2 in order to reduce the redundancy. - Line 52, 116: The authors mention complex karyotype as a common factor associated with poor prognosis in RT. However, they do not specify how this term is defined. To improve the comprehensiveness of the article, the authors should briefly describe what a complex karyotype is, and how it has been defined.
Response: Thank you for this helpful suggestion. We have clarified the definition of complex karyotype in the manuscript.
Lines 49 to 54: The revised text: RT is predisposed by both clinical and biological high-risk features: lymphadenopathy >5 cm, Rai stage III–IV, unmutated immunoglobulin heavy chain variable region (IGHV) status, TP53 aberrations, NOTCH1 mutations, presence of stereotyped B-cell receptors, over-expression of CD38 or ZAP-70, and a complex karyotype, commonly defined as the presence of ≥3 numeric or structural chromosomal abnormalities by conventional karyotyping. - Line 171: The authors mention minimal residual disease (MRD), a key concept for the assessment of responses and potentially cure in CLL and RT. This aspect is not further followed up on. The authors should add some additional explanation about how MRD is assessed in RT, whether it has been successfully used in clinical trials, and where its future role in RT treatment might be. Some of that is implied in table 1 and section 6, but it should be presented more clearly in one place.
Response: Thank you for this insightful comment. We have revised the manuscript to include a dedicated paragraph on MRD in RT. We now clarify how MRD is assessed (flow cytometry and next-generation sequencing), We also note the potential future application of MRD to guide consolidation strategies and trial design.
The added text:
Lines 141 to 148: MRD is a validated prognostic tool in CLL, typically measured by multicolor flow cytometry (~10⁻⁴ sensitivity) or next-generation sequencing (~10⁻⁶ ) [46, 47]. Its prognostic role in RT remains undefined and should be interpreted cautiously. Recent CAR T-cell therapy studies in RT report high response rates but have not consistently integrated MRD assessment into outcome measures CLL [48, 49]. While MRD negativity pre allo HSCT correlates with superior outcomes in aggressive lymphoma cohorts, specific data in RT are lacking[49, 50]. Prospective RT trials incorporating MRD as a biomarker to guide consolidation strategies would be worthwhile.
- Line 409: ”Novel immune cell engagers […] preclinical evaluation”. This statement should be backed up with a specific reference.
Response: Thank you for pointing out the error. We have added 3 references (124 to 126) in order to support this statement in line 506.
- Line 417: “selective CDK9 […] downregulating MCL1”. This statement should be backed up with a specific reference.
Response: Thank you for your suggestion. We have added reference number 126 in order to support this statement.

Reviewer 3 Report
Comments and Suggestions for Authors
The perspectives for patients with Richter transformation (RT) of chronic lymphocytic leukemia (CLL) have significantly evolved with the advent of novel therapies. Over the past year, several high-quality and thematically similar review articles have been published on this topic, including:
Barrett M. Richter's transformation: Transforming the clinical landscape. Blood Reviews. March 2024;64:101163.
Rippel N, Sheppard R, Kittai AS. Updates in the Management of Richter Transformation. Cancers. 2025;17(1):95. https://doi.org/10.3390/cancers17010095
Thompson PA, Parry EM. Advances in the Understanding of Richter Transformation. Hematol Oncol Clin North Am. Published online July 7, 2025. doi:10.1016/j.hoc.2025.05.011
Sander B. Richter transformation recommendations. Blood. 2025;146(3):262–263. doi:10.1182/blood.2025029450
Given the timeliness and thorough coverage of the topic in recent literature, it is essential to modernize and enrich the current manuscript to ensure it offers added value and distinctiveness. Specifically, I recommend incorporating recent clinical experiences and trial results involving immune checkpoint inhibitors (e.g., Tedeschi A et al., Lancet Oncology 2024), copanlisib plus nivolumab (Shouse G et al., Haematologica 2025), CD19-directed CAR-T cell therapy (e.g., Nadiminti KV et al., Transplant Cell Ther 2025), as well as ongoing trials such as the Euplagia-1 study (Ortiz-Maldonado V et al., Blood 2024) and blinatumomab trials (Guièze R et al., Nat Commun 2024).
In addition to updating the therapeutic section, I strongly advise expanding Section 2: Molecular Features and Pathogenesis with a more detailed explanation of the molecular mechanisms underlying Richter transformation. This should include the most relevant genetic alterations nd their clinical and therapeutic implications. I also recommend including a graphical illustration (e.g., a schematic showing the clonal evolution from CLL to RT with associated molecular changes), which would enhance both clarity and educational value for readers.
Author Response
Reviewer 3
The perspectives for patients with Richter transformation (RT) of chronic lymphocytic leukemia (CLL) have significantly evolved with the advent of novel therapies. Over the past year, several high-quality and thematically similar review articles have been published on this topic, including:
Barrett M. Richter's transformation: Transforming the clinical landscape. Blood Reviews. March 2024;64:101163.
Rippel N, Sheppard R, Kittai AS. Updates in the Management of Richter Transformation. Cancers. 2025;17(1):95. https://doi.org/10.3390/cancers17010095
Thompson PA, Parry EM. Advances in the Understanding of Richter Transformation. Hematol Oncol Clin North Am. Published online July 7, 2025. doi:10.1016/j.hoc.2025.05.011
Sander B. Richter transformation recommendations. Blood. 2025;146(3):262–263. doi:10.1182/blood.2025029450
Given the timeliness and thorough coverage of the topic in recent literature, it is essential to modernize and enrich the current manuscript to ensure it offers added value and distinctiveness. Specifically, I recommend incorporating recent clinical experiences and trial results involving immune checkpoint inhibitors (e.g., Tedeschi A et al., Lancet Oncology 2024), copanlisib plus nivolumab (Shouse G et al., Haematologica 2025), CD19-directed CAR-T cell therapy (e.g., Nadiminti KV et al., Transplant Cell Ther 2025), as well as ongoing trials such as the Euplagia-1 study (Ortiz-Maldonado V et al., Blood 2024) and blinatumomab trials (Guièze R et al., Nat Commun 2024).
In addition to updating the therapeutic section, I strongly advise expanding Section 2: Molecular Features and Pathogenesis with a more detailed explanation of the molecular mechanisms underlying Richter transformation. This should include the most relevant genetic alterations nd their clinical and therapeutic implications. I also recommend including a graphical illustration (e.g., a schematic showing the clonal evolution from CLL to RT with associated molecular changes), which would enhance both clarity and educational value for readers.
Response: Thank you for this important suggestion. We have modernized the manuscript to incorporate recent clinical experiences and trial data. Specifically, we have done following modification:
Section 2 (Molecular Features): Expanded discussion of genetic alterations (TP53, NOTCH1, CDKN2A/B, MYC) and their clinical/therapeutic implications. We also added a schematic figure 1 illustrating the clonal evolution from CLL to RT. Owing to word limit constraints, this section was kept concise, but it highlights the most relevant mechanisms underlying transformation.
Therapeutic sections: Updated with recent trials, including immune checkpoint inhibitors (reference 117: Tedeschi 2024; reference 118: Shouse 2025), CD19-directed CAR T-cell therapy (Reference 103: Nadiminti 2025), and novel immunotherapies such as Euplagia-1 (Reference 115: Ortiz-Maldonado 2024) and blinatumomab (Reference 33: Guièze 2024).
These additions enrich the review with up-to-date clinical data, mechanistic insights, and visual context, enhancing both educational value and distinctiveness compared with existing literature.

Round 2
Reviewer 1 Report
Comments and Suggestions for Authors
Thank you for this revised version that addressed all points previously raised.
Author Response
We sincerely thank the reviewer for the positive feedback and for acknowledging our revisions. We are pleased that the current version has satisfactorily addressed the previous concerns.
Reviewer 2 Report
Comments and Suggestions for Authors
Summary:
In the presented revised manuscript titled “Richter Transformation in Chronic Lymphocytic Leukemia: Current Treatment Challenges and Evolving Therapies”, Lin et al. summarize the biology, current and future treatments, and challenges in the management of Richter transformation (RT). The manuscript is a review article where the authors first summarize the pathologic and molecular features of RT, including its clonal origin, subtypes, major genomic features and overall prognosis. Aspects surrounding biomarkers and minimal residual disease (MRD), as well as hypotheses regarding the link between RT risk and different chronic lymphocytic leukemia (CLL) therapeutic regimens are discussed. The authors then go on to describe the available treatments for RT and their respective prognosis, including BTK inhibitors, BCL2 inhibitors, allogeneic hematopoietic stem cell transplant (Allo-HSCT), bispecific antibodies and chimeric antigen receptor (CAR)-T cell therapy. The authors outline risks and benefits of these treatments, and ongoing/completed clinical trials. They further highlight future therapies under investigation. The article closes with a summary of remaining challenges for the treatment of RT, such as limited duration of treatment response, and the prevalent frailty/morbidity of RT patients.
General Concept comments:
The article is concisely written and clearly structured. It summarizes the evolving field of RT and RT treatments well and cites many recent studies in the field. It delivers clear summaries of treatment concepts for RT in several tables and one figure. The conclusions and discussion points highlighted by the authors are fully justified by the data/studies presented.
In the revised manuscript, the authors have addressed reviewer comments and added in several sections that now outline additional topics surrounding RT. This includes additional discussion of the risk factors of RT and minimal residual disease (section 3 in the manuscript) and an additional table (table 1) summarizing RT incidence in the literature. Differences in incidence and its hypothesized links to therapy are discussed. This is being supplemented by a new figure (Figure 1) schematically illustrating the clonal evolution from CLL to RT and associated therapeutic implications.
The previous table 2 has been removed and information incorporated into the manuscript text. Several terms that were left unclear previously are now defined more specifically in the text, including minimal residual disease (MRD), the definition of complex karyotypes and the definition of complete response. Several literature references were added to back up the statements made in the article. Additional discussion has also been added regarding CAR-T cells, especially CAR-T cell toxicities (section 6), and checkpoint inhibitor combinations (section 8).
Overall, the authors have addressed multiple comments regarding organization/layout and content of the manuscript. They have significantly improved the comprehensiveness of the review by incorporating additional topics. They further consolidated tables and reorganized literature data presentation (table 1), making the manuscript more organized. The resulting manuscript has thus fully satisfied the reviewer requests and criticisms.
Author Response
We sincerely thank the reviewer for the thorough evaluation and the positive feedback on our revised manuscript. We greatly appreciate your recognition of the added discussions on risk factors, MRD, therapeutic implications, and the expanded content on CAR-T toxicities and checkpoint inhibitor combinations. Your constructive comments have been instrumental in improving the comprehensiveness, clarity, and overall organization of our review. We are pleased that the revisions have fully satisfied your requests and criticisms, and we thank you again for your thoughtful insights that helped strengthen the manuscript.
Reviewer 3 Report
Comments and Suggestions for Authors
The authors have only partially addressed the questions raised. Section 2 has been slightly expanded, and the authors stated that this was “owing to word limit constraints.” To the best of my knowledge, there are no strict word limits for IJMS, and in my opinion, it is essential to provide a more detailed and scientifically rigorous description of the molecular mechanisms underlying Richter's transformation (RT), which aligns fully with the scope of this journal.
This is particularly important given the clonal relationship with the underlying CLL and the molecular characteristics (presumably, but not limited to TP53 status), which have clear clinical and therapeutic implications.
If the authors are indeed concerned about “word limit constraints,” it is unclear why they duplicated the same statement in Section 2 — Lines 67–69: “The most recurrent alterations include TP53 mutations and deletions, observed in 60–70% of RT cases, which lead to impaired DNA damage response and resistance to chemoimmunotherapy [21]. CDKN2A/B loss promotes unchecked cell-cycle progression…” — and again in Lines 87–89: “Genetic markers like TP53 mutations (seen in 60 to 70% of RT cases) and CDKN2A loss are frequent [34].”
The added Figure 1 is uninformative, as the underlying molecular mechanisms are not presented.
Author Response
Comment 1:
The authors have only partially addressed the questions raised. Section 2 has been slightly expanded, and the authors stated that this was “owing to word limit constraints.” To the best of my knowledge, there are no strict word limits for IJMS, and in my opinion, it is essential to provide a more detailed and scientifically rigorous description of the molecular mechanisms underlying Richter's transformation (RT), which aligns fully with the scope of this journal.
This is particularly important given the clonal relationship with the underlying CLL and the molecular characteristics (presumably, but not limited to TP53 status), which have clear clinical and therapeutic implications.
If the authors are indeed concerned about “word limit constraints,” it is unclear why they duplicated the same statement in Section 2 — Lines 67–69: “The most recurrent alterations include TP53 mutations and deletions, observed in 60–70% of RT cases, which lead to impaired DNA damage response and resistance to chemoimmunotherapy [21]. CDKN2A/B loss promotes unchecked cell-cycle progression…” — and again in Lines 87–89: “Genetic markers like TP53 mutations (seen in 60 to 70% of RT cases) and CDKN2A loss are frequent [34].”
The added Figure 1 is uninformative, as the underlying molecular mechanisms are not presented.
Response:
We thank the reviewer for the critical and constructive comments, which have guided us in substantially improving Section 2. In the revised manuscript, we have now significantly expanded the section on Molecular Features and Pathogenesis to provide a more detailed and scientifically rigorous description of the molecular mechanisms underlying Richter transformation (RT).
Specifically, the new Section 2 now:
-
Elaborates on the clonal relationship between CLL and RT, emphasizing the evolutionary continuum and the “early seeding” model supported by longitudinal studies.
-
Provides in-depth mechanistic discussion of the four principal driver lesions—TP53 inactivation, CDKN2A/B loss, NOTCH1 activation, and MYC dysregulation—and explains how these alterations converge with chronic BCR signaling to drive transformation.
-
Discusses the synergy between tumor suppressor disruption and antigen-driven signaling, metabolic reprogramming, and genomic instability (chromothripsis, chromoplexy, whole-genome doubling).
-
Highlights immunogenetic factors such as unmutated IGHV and stereotyped BCRs (e.g., subset #8), as well as the activated B-cell phenotype and immune evasion features of RT.
-
Identifies outstanding gaps, including the temporal ordering of driver events, therapy-driven evolutionary pressures, and limitations of current models.
We have also corrected the redundancy previously noted regarding TP53/CDKN2A statements, ensuring clarity and avoiding repetition.
In addition, we have replaced the prior schematic with a new, more informative figure (Figure 1) that integrates the recurrent genetic lesions (TP53, CDKN2A/B, NOTCH1, MYC) with BCR signaling and downstream pathways (PI3K–AKT, NF-κB). The figure and its revised legend explicitly illustrate how these alterations converge to drive clonal evolution and the emergence of aggressive RT clones.
We are grateful for the reviewer’s feedback, which has enabled us to align the manuscript more closely with the journal’s scope and to enhance both scientific rigor and educational value.
